# Wear Analysis of Four Different Single-File Reciprocating Instruments before and after Four Uses in Simulated Root Canals

Alessio Zanza [1] , Rodolfo Reda [1] , Giuseppe Familiari [2,*], Stefania Annarita Nottola [2], Dario Di Nardo [1] , Luca Testarelli [1] and Orlando Donfrancesco [1,2]

1 Department of Oral and Maxillo-Facial Science, Sapienza University of Rome, Via Caserta 06, 00161 Rome, Italy; alessio.zanza@uniroma1.it (A.Z.); rodolfo.reda@uniroma1.it (R.R.); dario.dinardo@uniroma1.it (D.D.N.); luca.testarelli@uniroma1.it (L.T.); orlando.donfrancesco@uniroma1.it (O.D.)
2 Department of Anatomical, Histological, Forensic and Orthopaedic Sciences, Section of Human Anatomy, Sapienza University of Rome, Via A. Borelli 50, 00161 Rome, Italy; stefania.nottola@uniroma1.it
* Correspondence: giuseppe.familiari@uniroma1.it

**Abstract:** The aim of this study is to assess the surface alterations of four reciprocating instruments before and after the shaping of four resin-simulated root canals. The following four different reciprocating instruments are selected: 10 Reciproc Blue (RB25), 10 WaveOne Gold (WOG), 10 EdgeOne Fire (EOF) and 10 recently introduced instruments OneRECI (OR), for a total of 40 new instruments. Before root canal shaping, each instrument is mounted on a stub in a standardized position and observed using a scanning electron microscope (SEM) to detect any surface alteration, such as microfractures, metal defects, deformations, blunt and disruption of cutting edges, debris, pitting and tip flattening. Micrographs are acquired at the level of the tip, 4 mm, 8 mm and 12 mm from it. After that, each instrument is used in four simulated resin root canals. SEM observation was repeated after the simulated clinical use to assess the wear resistance of the instruments. Surface alterations are registered before and after instrumentation and statistical analysis is performed using a Chi-Square test to verify homogeneity of defects distribution and GLM to evaluate the differences of RMS at baseline and after use for both groups ($\alpha$ level 0.05). Before simulated clinical uses, no alterations are found except for three cases of EOF with metal strips in correspondence to the tip, disruption of the cutting edge of WOG and oily spotting on two different OR. After simulated clinical use, EOF shows a statistically significant difference in terms of spiral distortion and flattening of the cutting edges. The OR shows the highest presence of debris despite the ultrasonic cleaning procedures. No instrument fractures are observed. EOF should be discarded after four clinical uses and carefully inspected after each insertion into root canals. Moreover, tough attention should be paid during disinfection and cleaning procedures after instrumentation considering the copious debris detected in each instrument, particularly in OR.

**Keywords:** endodontics; nickel-titanium rotary instruments; reciprocation; root canal treatment; scanning electron microscopy; wear analysis

## 1. Introduction

The reciprocating motion used with nickel-titanium rotary instruments was introduced in the early 2000s, revolutionizing the kinematics in endodontics, despite the concept of reciprocation being already validated on stainless-steel manual files since the mid-1990s [1,2]. This motion is defined as a rotation of a determined angle in the cutting verse followed by a second rotation, more frequently of a lower angle, in the opposite noncutting verse. This alternating rotation determines the final rotation of the endodontic instrument that will make a complete rotation around its axis for a certain number of

reciprocating cycles according to the difference between the angle in the cutting verse and the angle in the noncutting one [1,3]. The benefits arising from this kinematic have been thoroughly investigated, concluding that the reciprocating motion is able to increase both cyclic fatigue and torsional resistance. Regarding the first one, it has been demonstrated that reciprocating movements increase the cyclic fatigue resistance in comparison to continuous rotation. This is due to the fact that the alternating rotation leads to a distribution of the tension–compression strain cycles acting on the region of the instrument engaged in the point of the maximum curvature of the root canal, reducing the concentration of stresses [4]. Moreover, since the instruments used in reciprocation are subjected to less torsional stress since they continuously engage and disengage dentine, reducing the incidence of torsional fracture by taper-lock [5,6]. Despite this, over the years, several limitations have been highlighted. Firstly, reciprocating instruments have been criticized for the possibility of extruding debris and bacteria in the periapical space in comparison to continuous rotary instrumentation [3,7,8]. Despite this, to date, the literature includes conflicting results, so further studies are needed [3]. Nevertheless, during the use of single-file reciprocating instruments, it is suggested to recapitulate and irrigate after each insertion into the canal system in order to avoid any blockage and extrusion of debris [9]. Another point that has been raised is the less cutting efficiency of the instrument used with a reciprocating motion. Nevertheless, even in this case, the data are conflicting and a unanimous consensus was not reached [3].

The above-mentioned advantages allow the use of only one instrument to determine the final shape of the root canal system in a safe way without excessive risk of intracanal separation [1,3]. This could not be performed with complete rotation because of the risk of fracture, which is severely reduced with the reciprocation. Despite this, manufacturers suggest the single use of this type of instrument since all mechanical stresses arising from shaping are concentrated on a single instrument and not spread among more instruments in a sequence [3]. Regarding this, the knowledge of the mechanical resistance of reciprocating single-file and their wear resistance is fundamental to obtaining an in-depth comprehension of their mechanical limits, in order to avoid as much as possible their intracanal separation and using them as effective as possible, reducing the waste and discard of several instruments.

The most suitable method to analyze the wear resistance of endodontic instruments is through scanning electron microscopy (SEM), which allows the observation at higher magnification in order to detect any surface alteration such as microfractures, metal defects, deformations, blunt cutting edges, disruption of cutting edges, presence of debris, pitting and tip flattening [10,11].

According to this, the aim of this study was to assess and compare the wear resistance of two recently introduced single-file reciprocating instruments (OneRECI (Coltene, MicroMega, Besançon, France)) and EdgeOne Fire (EdgeEndo, Albuquerque, NM, USA) with two other well-known martensitic single-file reciprocating instruments Reciproc Blue (VDW, Munich, Germany) and WaveOne Gold (Dentsply Maillefer, Ballaigues, Switzerland) through an SEM observation by comparing surface defects before and after the simulated use of the instruments in resin lower molars.

## 2. Materials and Methods

### 2.1. Sample Size Calculation and Groups Definition

The sample size was calculated on the basis of previously published study using G*Power v3.1 (Heinrich Heine, University of Düsseldorf, Düsseldorf, Germany) by setting an alpha-type error of 0.05, a beta power of 0.90 and an effect size of 0.80 [12]. A total of 6 samples per group were indicated as the ideal size required for noting significant differences. However, additional 4 samples per group were added in order to compensate for unexpected values of OneRECI and EdgeOne Fire since there were no data in the literature regarding their wear resistance. According to this, a total of 10 instruments per group were selected. Thus, the following four groups were determined according to the

instrument brand: Reciproc Blue (RB group), WaveOne Gold (WOG group), EdgeOne Fire (EOF group) and OneRECI (OR group).

Instruments from each group were previously inspected through a stereomicroscope at 20× (Carl Zeiss Microimaging, Göttingen, Germany) to eventually evidence macroscopical defects and none of them was discarded.

### 2.2. Pre-Operative Surface Analysis

Fifteen brand new sterile instruments were mounted on a stub in a standardized position avoiding any contact with other materials and the consequent contamination of instrument's surface. Micrographs were acquired with a high-pressure scanning electron microscope (VP-SEM; SU3550, Hitachi High Technologies Corporation, Tokyo, Japan) at a variable magnification ranging from 100× to 1000× at four different levels of each instrument: tip and 4 mm, 8 mm and 12 mm from it. After the establishment of the baseline, each instrument was carefully inspected by two different examinators along its entire length in order to detect any of the following surface alterations: microfractures, metal defects, deformations, blunt cutting edges, disruption of cutting edge, presence of debris, pitting and tip flattening. In case of the presence of those defects, additional photomicrographs were acquired, specifying the zone of the surface alteration taking as reference points the previous and the following cutting edges to reinspect the zone after the simulated clinical use. The above-mentioned procedure was repeated turning each instrument at 15° on both left and right sides around its longitudinal axis thanks to the possibility of rotation of the complex formed by the stub and the platform of the SEM.

### 2.3. Instrumentation of Resin Teeth

The instrumentation of root canal system was simulated by shaping resin models of lower molars (*n* = 60) (Fanta Dental, Shanghai, China) characterized by two distal canals and two mesial canals (Figure 1). The teeth were received with a performed access cavity. Each canal was characterized by an angle of curvature of 60° and a radius of curvature of 5 mm.

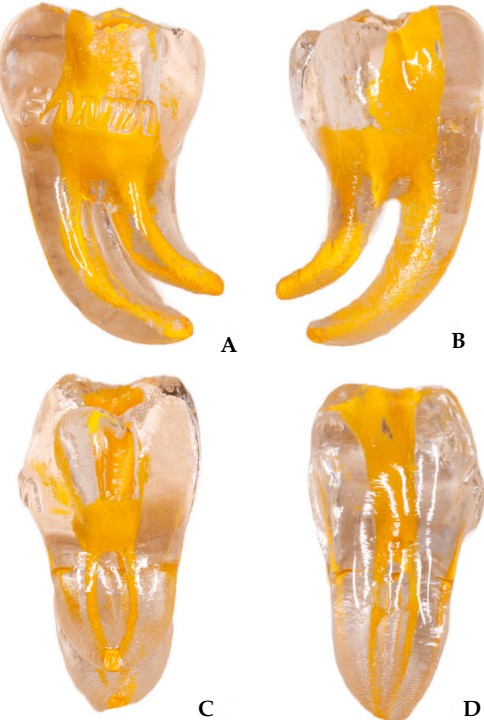

**Figure 1.** Resin lower molar sample characterized by 4 canals, 2 mesial and 2 distal of 60° of curvature with a radius of curvature of 5 mm. (**A**) vestibular aspect; (**B**) lingual aspect; (**C**) distal aspect; (**D**) mesial aspect.

Each canal was instrumented by a single expert operator according to the following procedure: Firstly, a manual glide path was performed with a K-file #10 (Dentsply Maillefer, Ballaigues, Switzerland) to ensure the patency of the canal; then, the entire shaping was performed with a single-file reciprocating technique using the selected instrument according to the manufacturer's recommendation. According to this, RB was used in a reciprocating mode with the cutting angle set to 150° and a non-cutting angle set to 30°, WOG and EOF were used in a reciprocating mode with the cutting angle set to 170° and a non-cutting angle set to 50°, whilst OR were used in a reciprocating mode with the cutting angle set to 170° and a non-cutting angle set to 60°. The rotational speed was set to 300 rpm for each instrument.

The progression to the working length was reached with a slow in-and-out pecking motion without any brushing action. The inward movement was 2 mm, whilst the outward movement was 1 mm. After three consecutive cycles of inward and outward movements the instrument was removed from the canal, cleaned and the canal was irrigated with NaOCl solution at 5.25%, which was activated with a sonic device (Endoactivator, (Dentsply Maillefer, Ballaigues, Switzerland)) for 20 s. The patency was checked with K-file #10 before each re-insertion of the instrument.

### 2.4. Post-Operative Surface Analysis

The used instruments were mounted on the stub in the standardized position after being cleaned in an ultrasonic bath for 10 min in order to remove debris accumulated consequently to the instrumentation. The observation procedures were performed with the same protocol as the pre-operative surface analysis.

### 2.5. Statistical Analysis

Chi-square test has been used as a homogeneity test for evaluating the deviation of distribution of the identified defects with respect to the estimated defects. The differences of RMS at baseline and after uses for the four single-file reciprocating instruments were compared performing GLM model for repeated measures; $\alpha$ level was a priori set as 0.05.

### 3. Results

All results regarding the incidence of pre-operative and post-operative surface defects are shown in Table 1.

**Table 1.** Schematic representation of the incidence of pre-operative (green cells) and post-operative (white cells) surface defects in relation to the selected instruments. EOF (EdgeOne Fire), WOG (WaveOne Gold), R25 Blue (Reciproc Blue).

| | EOF (n = 10) | | | | WOG (n = 10) | | | | R25 Blue (n = 10) | | | | OneRECI (n = 10) | | | |
|---|---|---|---|---|---|---|---|---|---|---|---|---|---|---|---|---|
| | Tip | 4 mm | 8 mm | 12 mm | Tip | 4 mm | 8 mm | 12 mm | Tip | 4 mm | 8 mm | 12 mm | Tip | 4 mm | 8 mm | 12 mm |
| Microfractures | 0 0 | 0 6 | 0 2 | 0 0 | 0 0 | 0 3 | 0 1 | 0 0 | 0 0 | 0 2 | 0 1 | 0 0 | 0 0 | 0 3 | 0 1 | 0 0 |
| Metal defects | 3 4 | 0 0 | 0 2 | 0 0 | 0 0 | 0 0 | 0 0 | 0 0 | 0 0 | 0 0 | 0 0 | 0 0 | 0 0 | 0 0 | 0 0 | 0 0 |
| Blunt cutting edges | 0 7 | 0 6 | 0 8 | 0 8 | 0 2 | 0 3 | 0 2 | 0 2 | 0 0 | 0 1 | 0 2 | 0 1 | 0 1 | 0 2 | 0 1 | 0 1 |
| Disruption of cutting edge | 0 0 | 0 0 | 0 3 | 0 0 | 10 10 | 10 10 | 10 10 | 10 10 | 2 3 | 0 2 | 0 3 | 0 4 | 0 1 | 0 2 | 0 0 | 0 2 |
| Debris | 3 2 | 4 2 | 3 1 | 3 2 | 0 0 | 1 0 | 1 3 | 1 2 | 0 5 | 3 5 | 2 5 | 1 2 | 1 3 | 1 5 | 1 8 | 1 10 |
| Pitting | 0 2 | 4 6 | 0 3 | 0 1 | 0 2 | 0 2 | 0 2 | 0 1 | 0 2 | 0 2 | 0 3 | 0 0 | 0 4 | 0 4 | 0 3 | 0 1 |
| Tip flattening | 0 | | 8 | | 0 | | 2 | | 0 | | 2 | | 0 | | 3 | |

Any surface alteration was detected in all as-received instruments except for 3 of the 10 inspected EOF, in which metal defects probably arose from the machining procedures were detected at the instrument's tip, with a statistically significant difference in comparison to the other brands ($p < 0.05$) (Figure 2). Nevertheless, after four uses of the instruments,

those points of surface alteration did not act as points of least resistance for the development of microfractures.

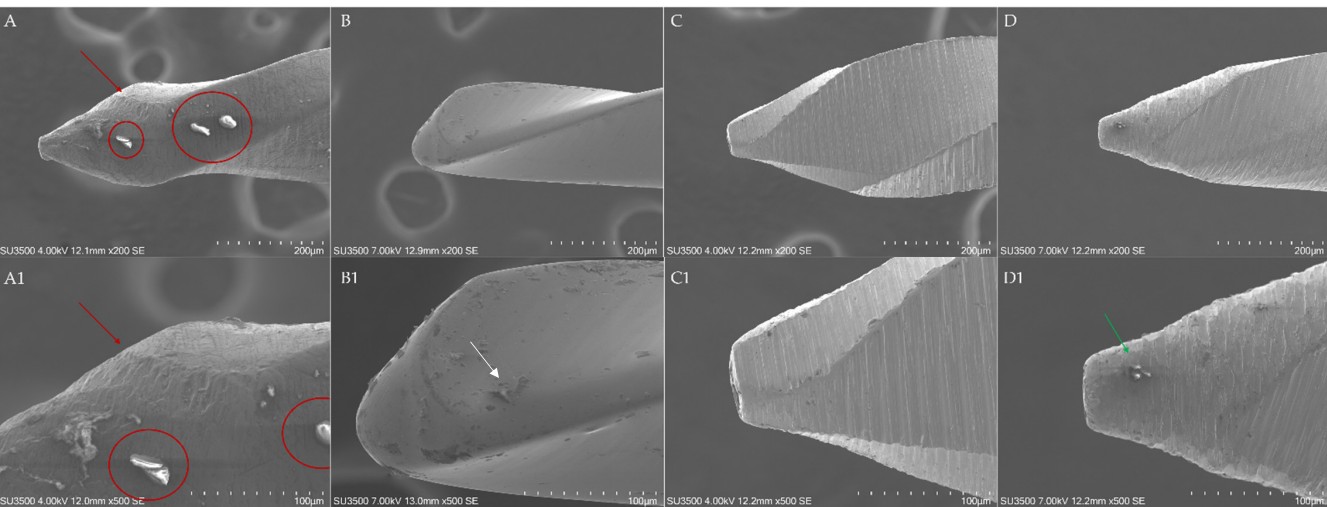

**Figure 2.** Sem images of tips of as-received four brand instruments. On the upper line are represented micrographs of the four instruments tips acquired at 200× magnification, whilst on the bottom line are represented micrographs at ×500 magnification. (**A,A1**) EOF tip, in which are shown metal defects such as microcavities and grooves (red arrows) and metal debris (red circle) respectively at ×200 and ×500 magnification. (**B,B1**) OR tip, in which are shown a smoother surface arising from proprietary surface treatment, at higher magnification (×500) some small debris can be noted (white arrows). (**C,C1**) WOG tip, in which are shown at different magnification (respectively, ×200 and ×500) the typical surface texture characterized by regular marks perpendicular to the longitudinal axis of the instrument. No debris and defects are noted. (**D,D1**) R25 Blue tip, in which are shown at different magnification (respectively ×200 and ×500) the typical surface texture characterized by regular marks perpendicular to the longitudinal axis of the instrument. Small debris are noted (green arrow).

Moreover, as-received EOF showed a statistically significant presence ($p < 0.05$) of metal debris arising from machining procedures along with the entire instrument in several cases, especially in the zone at 4 mm from the tip, with an incidence of 4 samples (Figure 3). The other brand instruments showed a limited and occasional (incidence of 0 or 1) presence of debris along their entire length, except for 3 samples of R25 Blue, in which debris were reported at 4 mm from the tip.

All WOG instruments showed a disruption of cutting edges for their entire length, maintained also after the shaping procedures (incidence of 10) (Figure 4). Those alterations did not act as points of least resistance for the development of microfractures. Moreover, the limited flattening of the cutting edges arising from simulated clinical use leads to a consequent reduction of cutting-edge disruption. It can be speculated that those morphological aspects arise from the machining procedures and do not have any role in the microcracks and fracture propagation.

Moreover, a spotty oil was found in two different OR instruments (Figure 3D).

During the shaping of resin root canals, no instrument fractures occurred, but a statistically significant difference was found in the detection of surface defects. Firstly, 3 EOF showed distortion of 1 or more spirals (stretched or shortened), localized in the zone ranging from 2 mm to 6 mm from the tip (Figure 5), with a statistically significant difference in comparison to the other brands ($p < 0.05$).

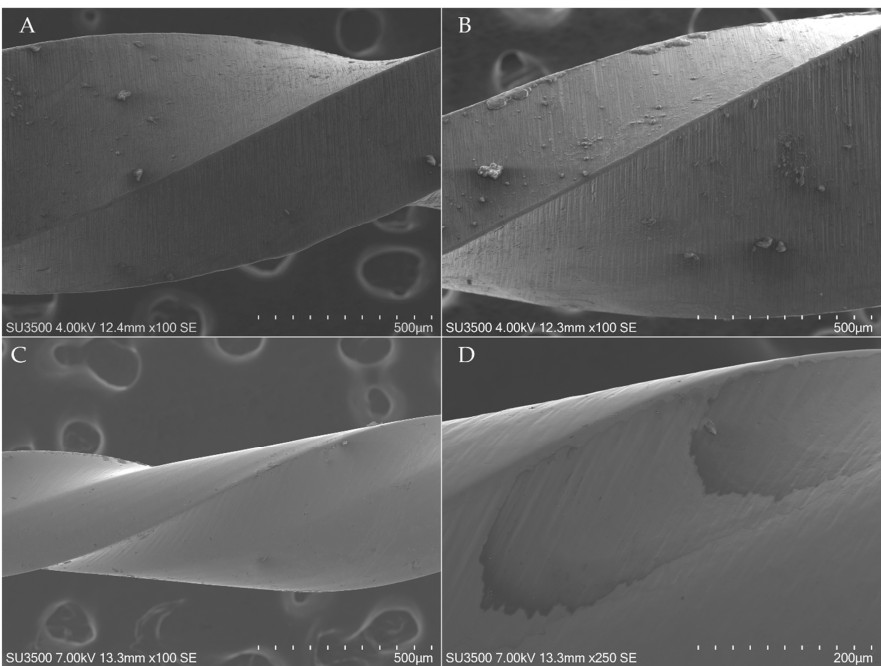

**Figure 3.** SEM images of debris on instruments surface. (**A**,**B**) EOF instruments, in which are shown several debris on the entire surface of the instrument. (**C**) OR instrument, in which are shown sporadic debris on the surface. (**D**) OR instrument, in which is shown an oily spot on the surface.

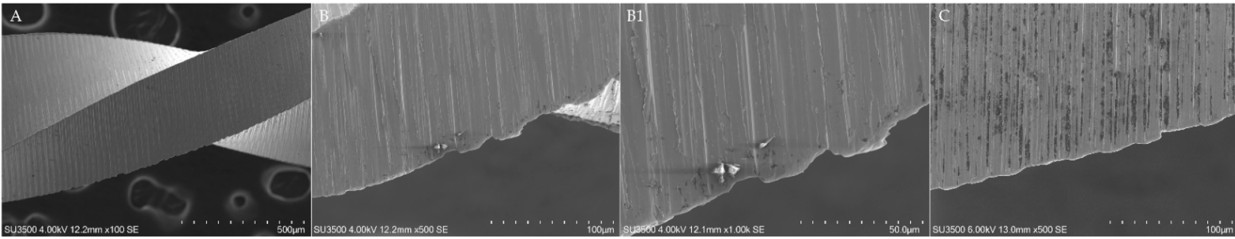

**Figure 4.** SEM images of the disruption of cutting edges in WOG new and used instruments. (**A**) ×100 magnification of the disrupted cutting edges in a new WOG instrument. (**B**,**B1**) two different magnifications (respectively ×500 and ×1000), in which is shown the presence of disruption of cutting edges without any associated microcracks or microfracture in WOG new instrument. (**C**) Micrograph showing the disruption of cutting edge in the used WOG instrument of image (**B**,**B1**), in which there has been the flattening of the cutting edges with a consequent reduction of their disruption. No microcracks or microfracture can be noted. It can be speculated that those morphological aspects arise from the machining procedures and do not have any role in the microcracks and fracture propagation.

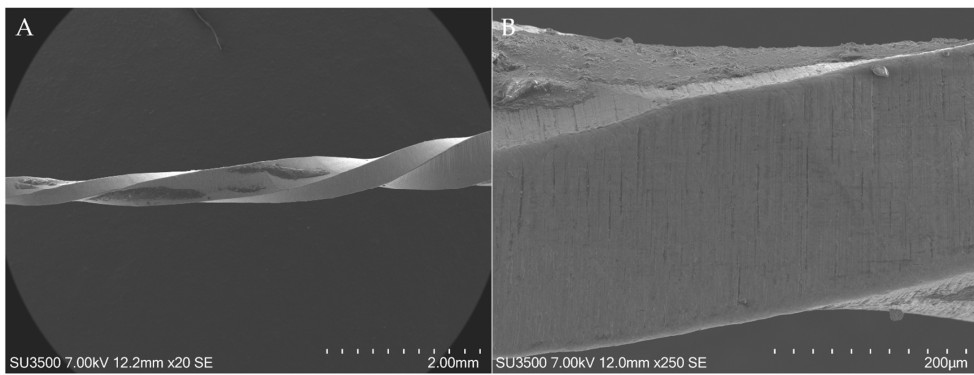

**Figure 5.** SEM micrographs showing the distortion of spirals of EOF instrument. (**A**) Stretching of the spiral at ×20 magnification. (**B**) Particular of image A acquired at higher magnification (×250), in which an alternation of the spiral can be noted.

Moreover, EOF showed a statistically significant flattening of the tip and cutting edges in comparison to other instruments (Figures 6–9), with an incidence of 8 and 7, 6, 8 and 8, respectively (respectively, in the tip and 4, 8 and 12 mm from it).

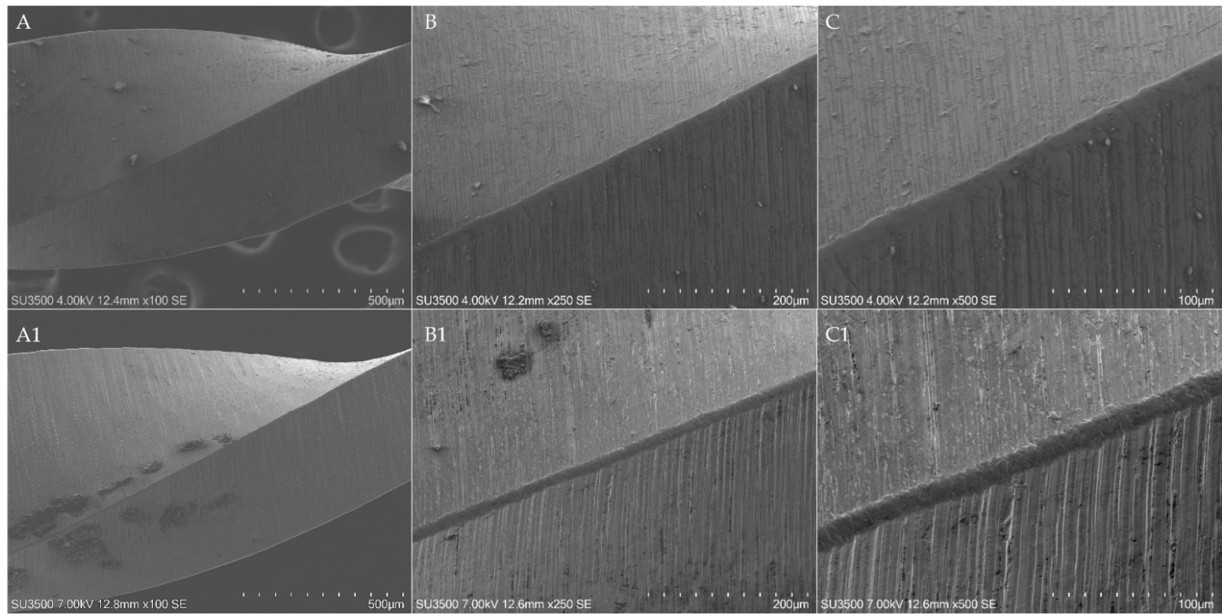

**Figure 6.** SEM images representing the flattening of cutting edges of EOF instrument after 4 simulated clinical uses at three different magnifications (from left to right respectively of ×100, ×250 and ×500). (**A–C**) cutting edge of EOF new instrument. (**A1–C1**) the same cutting edge of the EOF instrument acquired after 4 simulated clinical uses.

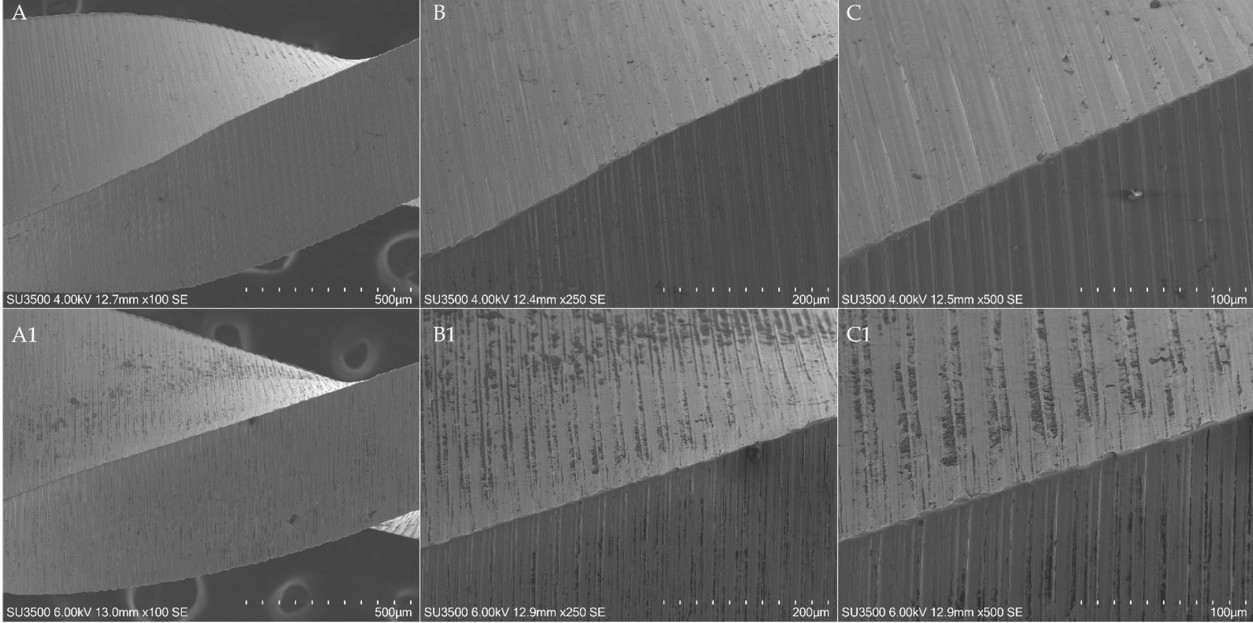

**Figure 7.** SEM images representing the limited flattening of cutting edges of WOG instrument after 4 simulated clinical uses at three different magnifications (from left to right respectively of ×100, ×250 and ×500). (**A–C**) cutting edge of WOG new instrument. (**A1–C1**) the same cutting edge of the WOG instrument acquired after 4 simulated clinical uses. A slight flattening can be appreciated only at ×500 magnification (**C,C1**).

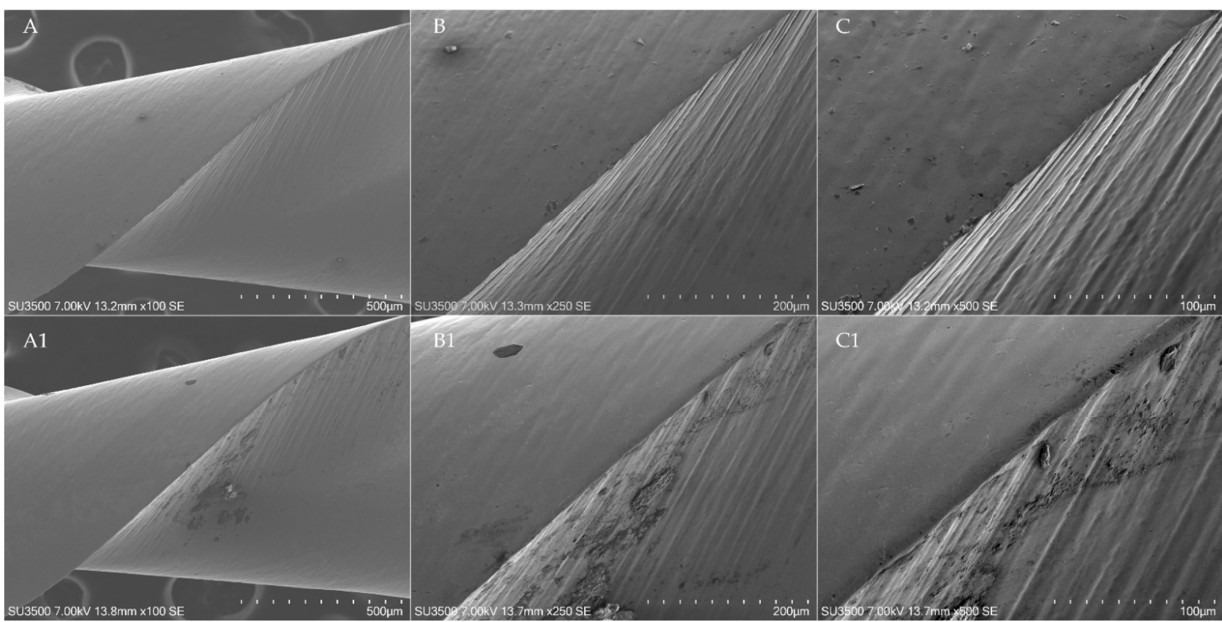

**Figure 8.** SEM images representing the moderate flattening of cutting edges of OR instrument after 4 simulated clinical uses at three different magnifications (from left to right respectively, of ×100, ×250 and ×500). (**A–C**) cutting edge of OR new instrument. (**A1–C1**) the same cutting edge of the OR instrument acquired after 4 simulated clinical uses. A moderate flattening can be appreciated at ×250 and ×500 magnification (**B,B1,C,C1**).

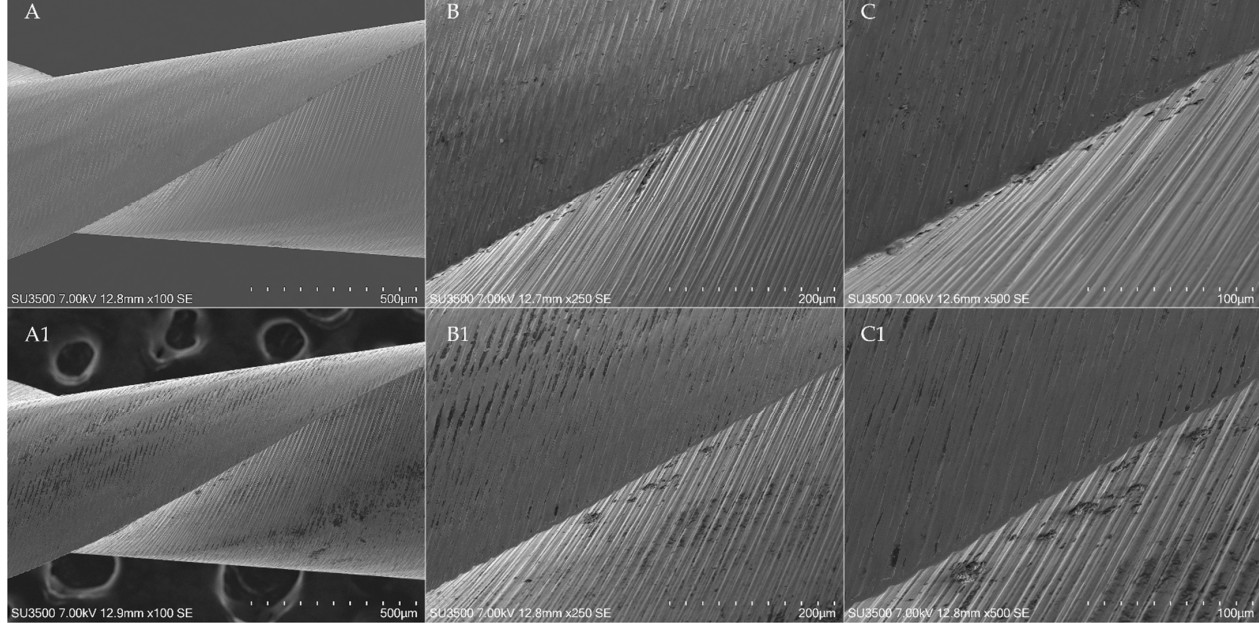

**Figure 9.** SEM images representing the imperceptible flattening of cutting edges of R25 Blue instrument after 4 simulated clinical uses at three different magnifications (from left to right respectively, of ×100, ×250 and ×500). (**A–C**) cutting edge of R25 Blue new instrument. (**A1–C1**) the same cutting edge of the R25 Blue instrument acquired after 4 simulated clinical uses.

Surface alterations such as microfractures and pitting were equally found in all brands, with no statistically significant differences among them ($p > 0.05$) (Figure 10).

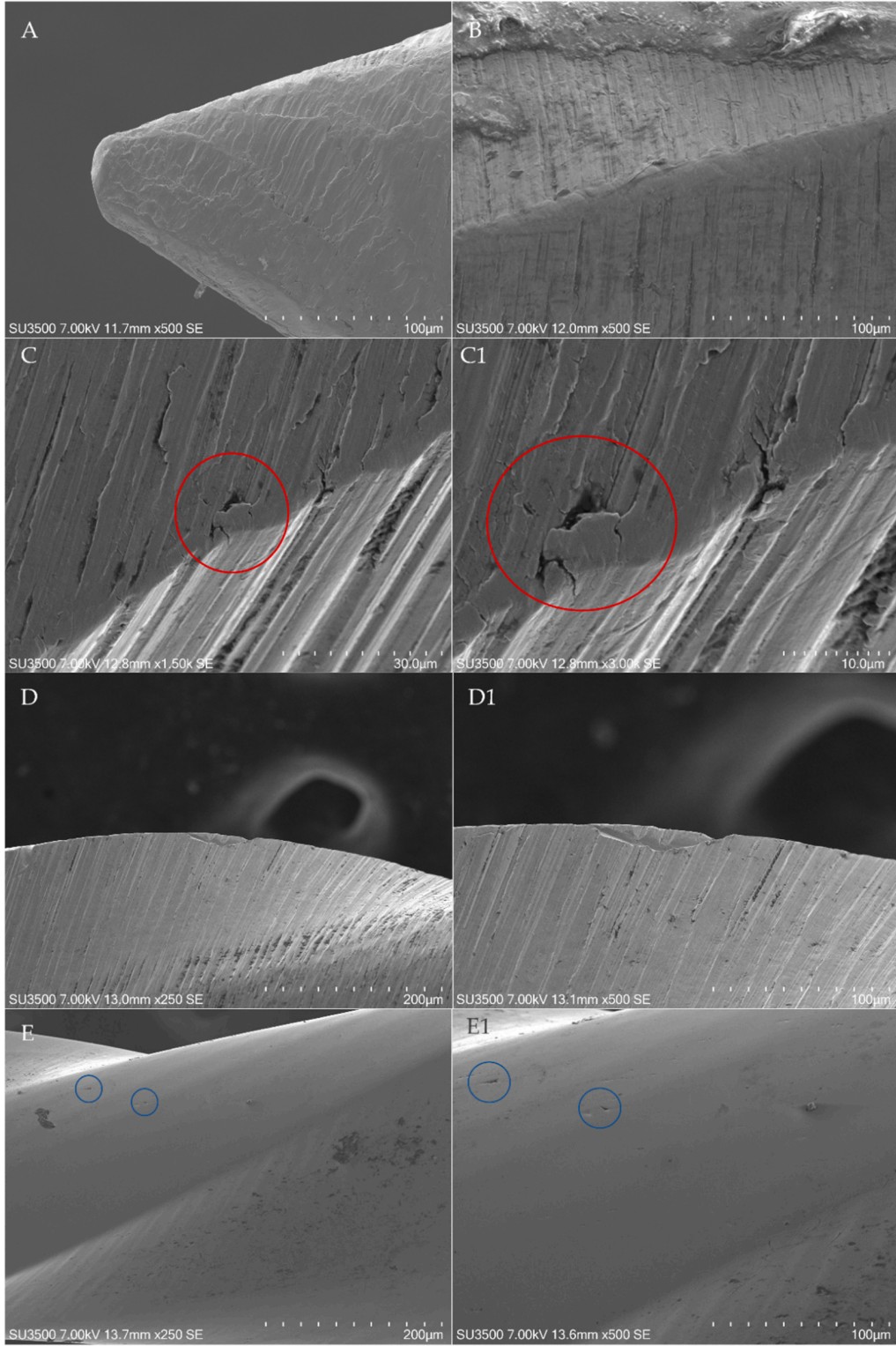

**Figure 10.** SEM micrographs of metal defects, microfractures and pitting of the 4 used brand instruments at different magnifications. (**A**) tip of a used EOF instrument, in which some grooves and metals defect can be noted at ×500 magnification. (**B**) EOF used instrument, in which some grooves and metals defect can be noted at ×500 magnification. (**C,C1**) microfractures (red circles) noted on the cutting edge of a used WOG instrument at two different magnifications, respectively, ×1500 and ×3000. (**D,D1**) sever disruption of the cutting edge of a used R25 Blue instrument with the association of microfractures (×250 and ×500 magnifications). (**E,E1**) pittings showed on a used OR surface at two different magnification ×250 and ×500 (blue circles).

Regarding the debris accumulated consequentially to the instrumentation, the OR showed the highest values ($p < 0.05$), with an incidence of 5 at the tip and at 4 and 8 mm from it and an incidence of 2 at 12 mm from the tip, followed by R25 Blue and then EOF and WOG.

## 4. Discussion

In the last decades, numerous technologies and improvements in endodontic instrumentation of root canal systems have been developed to achieve proper chemo-mechanical disinfection of root canals, avoiding as many as possible procedural errors and achieving predictable results in the shortest possible clinical time [3]. Recently, the reciprocating motion is gaining in popularity due to its positive clinical experience and scientific results. Undoubtedly, the introduction of single-file reciprocating systematics has facilitated the shaping procedures, simplifying the instrumentation protocols [1,3]. Despite this, the use of a single file for the entire procedure leads to the concentration of cyclic fatigue, flexural and torsional stresses on the single instrument, increasing its wear, despite the mechanical loads being severely reduced thanks to the reciprocating movements [13]. According to this, manufacturers strongly suggest the single use of those systematics, in order to reduce as much as possible their intracanal fracture, as well as the cross-contamination between patients. Nevertheless, this kind of approach leads to the waste of several instruments, increasing the actual cost of root canal treatments without the certainty that it is necessary to avoid instruments' intracanal failure. Regarding this, the wear analysis of reciprocating instruments is crucial to determine the possibility of safely reusing them after a single treatment. To date, there are few data on this topic and there are no data on the evaluation of wear resistance of WOG, R25 Blue, EOF and OR through SEM observation. Pirani et al. evaluated the difference in terms of wear resistance between WaveOne and Reciproc after three uses in narrow and straight canals, finding no instrument fractured and no macroscopic signs of plastic deformation, spiral distortion and limited surface defects such as pitting and microcracking [14]. Restrepo-Restrepo et al. evaluated the wear resistance of WOG and ProTaper Gold (PTG) (Dentsply Maillefer, Ballaigues, Switzerland) after clinical use, stating that the damage of the tip and cutting edge of WOG instruments (which underwent three instrumentation/sterilization cycles) was significantly more frequent than in PTG. According to the authors, the rationale for those findings is related to the different systematics between the two brands' instruments, with the stress concentrated on a single instrument in WOG [13].

According to the above-mentioned reasons, the aim of this study was to assess and compare the wear resistance of two recently introduced single-file reciprocating instruments, OR and EOF, with other two well-known martensitic single-file reciprocating instruments, R25 Blue and WOG, through SEM observation, by evaluating surface defects before and after the simulated use of the instruments in four canals of resin lower molars.

The surface characteristics of NiTi instruments have mainly been evaluated using SEM or atomic force microscopy (AFM) [11]. The first one is considered the most effective and diffused tool for the assessment of wear on instrument surface; however, some limitations have been raised, such as the inability to quantitatively assess the surface characteristics [15]. Despite this, the analysis of the wear defects could be greatly performed through a qualitative method since it consists of the evaluation of the presence or absence of defects. Despite many studies that have demonstrated the relationship between the increase of surface defects and the fracture of NiTi rotary instruments, the assessment of the relationship between the severity of defects and the probability of fracture is still unknown since they propagate slowly through the instruments until a sudden fracture occurs [13]. For those reasons, we believe that the benefits derived from the use of SEM for this aim are much greater than the drawbacks. Another limitation of the use of SEM is the risk of sample destruction caused by the preparation process [15]; however, the scanning electron microscope used in this research allowed the samples' examination without any preparation thanks to its functioning at high vacuum pressure.

As mentioned before, the knowledge of the wear resistance of endodontic instruments is fundamental to having an in-depth comprehension of their mechanical limits, in order to reduce as much as possible, the likelihood of intracanal fracture and using them as effective as possible, reducing the waste and discard of several instruments. According to this, to date, there are no data available regarding the comparison between the tested instruments.

The OR is a recently introduced single-file reciprocating instrument characterized by two different cross-section designs; a convex triangular for the apical 4 mm and an S-shaped for the rest of the instrument with a patented asymmetric cross-section. This manufacturing choice is probably intended to enhance the mechanical properties of the instrument, increasing the torsional resistance of the tip by increasing its polar moment of inertia [16] and increasing the cyclic fatigue resistance and the flexibility of the coronal part by reducing the cross-sectional mass with the S-shaped design [17]. Moreover, while the tip diameter is 0.25 mm, the taper is 0.06 variable and it is drastically reduced in the coronal part of the instrument thanks to the use of 1 mm wire diameter for its manufacturing, to preserve peri-cervical dentin during the instrumentation procedures [18]. The proprietary heat treatment of the alloy is the C-Wire.

The EOF is a recently introduced single-file reciprocating instrument conceived as a replica-like system of WOG with a different heat treatment, which, according to the manufacturer, should guarantee enhanced flexibility, cyclic fatigue and torsional resistance, despite the similar morphological characteristics such as cross-sectional design, tip diameter, taper, helix and pitch angle etc. [19]. However, Alcalde et al. comparing the morphological characteristics of EdgeTaper Platinum (replica-like systematic of PTG systematic) (EdgeEndo, Albuquerque, New Mexico) and PTG through a Micro-CT analysis found that, despite the manufacturer's claims, the metal mass volume ($mm^3$) and cross-section area ($\mu m^2$) were different with a statistically significant difference [20]. Those premises, in addition to the different heat treatments of the two instruments, could explain the differences in terms of mechanical and wear resistance between EOF and WOG.

According to the results, there were no statistically significant differences in the pre-operative surface analysis among the tested brand instruments, except for the incidence of metal defects and residual debris on EOF instruments, which probably arose from their manufacturing process. Despite this, the metal defects on the tips of the three EOF instruments did not interfere with their wear resistance after four canal uses since any microfracture or microcracks were found in those points in the post-operative analysis. The debris found on the EOF instruments before instrumentation are probably metal debris from wire blanks machining since their radiolucency in the SEM acquisition. After the instrumentation of 4 resin canals, the EOF showed the worst surface integrity in terms of unwinding, blunting of cutting edges and tip flattening. This denoted less wear resistance in comparison to other brands' instruments since they did not report statistically significant differences among them. The reason behind those findings could be the softness of the alloy arising from the different heat treatments that make the instruments more vulnerable to torsional stresses and friction against dentinal walls. The severe bluntness of the cutting edges strongly reduced the cutting efficiency of the EOF, probably making them unable to face the friction derived from the cutting action against resin, thus decreasing their torsional resistance.

Regarding the presence of debris after instrumentation and cleaning procedures, the OR showed the highest incidence, followed by R25 Blue. This is due to the similar cross-section shared by the two instruments, the S-Shaped design. This cross-sectional design significantly enhances the cutting efficiency and ensures a great removal of debris from the canals. Nevertheless, particular attention should be given to the disinfection protocols after their use in order to avoid any cross-contamination between patients. The difference in terms of debris on the surface of the instruments could derive from the depth of the flute of OR that is able to hold more debris.

The main limitation of this research is the use of simulated resin teeth instead of ex vivo or in vivo samples. This choice was made in order to standardize as much as possible

the external parameters influencing the wear resistance of instruments, such as the hardness of the cut material, the radius and the angle of curvatures, the canal diameter and the canal length since they have a significant influence on the mechanical resistance of NiTi endodontic instruments [21,22]. Selecting resin models, all those factors are considered constants, which is difficult with extracted teeth. Moreover, the resin teeth were fabricated without a roof chamber with a standardized access cavity, even though this parameter was constant among samples, avoiding any influences of different access cavities on the mechanical resistance of NiTi instruments [7,23]. Regarding this, most of the articles in the literature ignore the influence of all the above-mentioned factors on the wear resistance, thus increasing the risk of bias during the methodology. However, the results of this study should not be directly traduced in the clinical practice since the resin of tooth models has a different hardness in comparison to dentine. In fact, the aim of this study was not to assess the maximum number of instrumentable canals with a single instrument but to compare the wear resistance of different instruments. Despite this, the results obtained by Generali et al. regarding the evaluation of wear resistance of Reciproc Blue before and after four uses in extracted teeth with severe curvatures (ranging from 50° to 70°) are completely comparable with the results of this research [7]. The authors, evaluating ten samples for each instrument type, found the absence of unwinding, fractures and tip deformations, whilst four used R25 Blue instruments revealed microcracks along the surface perpendicular to the long axis [7]. Since the results of the instrument's surface defects arising from in vitro and ex vivo tests are comparable, it is possible to consider the methodology of this research reliable. Moreover, the use of simulated resin teeth could allow researchers to unconditionally increase the sample size, guaranteeing more reliable results without biological costs, as in the case of ex vivo studies, in which extracted teeth are required, also overcoming the difficulties in the sample selection. Moreover, the simulated resin teeth allow the control of variables such as curvature degrees, radius of curvature, canal diameter, canal length, hardness of the material and design of access cavity, guaranteeing more reliable results. Surely, further in vivo researches are needed to overcome those limitations.

## 5. Conclusions

According to the results and the limitation of the research, it can be concluded that particular attention should be given during the multiple uses of EOF instruments in comparison to other brands of instruments because of their reduced wear resistance that compromises their mechanical resistance and cutting efficiency, exposing them to an increased probability of intracanal failure. Moreover, in the case of multiple uses of single-file reciprocating instruments, thorough cleaning and disinfection are recommended, particularly with OR and R25 Blue.

**Author Contributions:** Conceptualization A.Z.; methodology, A.Z. and O.D.; software, R.R.; validation, A.Z., L.T. and G.F.; formal analysis, D.D.N.; investigation, A.Z. and O.D. resources, R.R.; data curation, L.T.; writing—original draft preparation, A.Z. and S.A.N.; writing—review and editing, O.D.; visualization, G.F.; supervision, G.F. and L.T.; project administration, G.F. All authors have read and agreed to the published version of the manuscript.

**Funding:** This research received no external funding.

**Institutional Review Board Statement:** Not applicable.

**Informed Consent Statement:** Not applicable.

**Data Availability Statement:** Not applicable.

**Conflicts of Interest:** The authors declare no conflict of interest.

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
