# Peer review of "Wear Analysis of Four Different Single-File Reciprocating Instruments before and after Four Uses in Simulated Root Canals"

_applsci, doi:10.3390/app12126253_

Round 1

Reviewer 1 Report

Thank you for the opportunity laid over on me to review this paper

The study is well executed and in the scope of the Journal

However the comments can be found below.

The abstract Line 14 should start with a capital word. Please correct the typographical error.

"in order to assess the surface alterations of four reciprocating instruments before and after shaping of four resin simulated root canals. " the sentence doesn't make sense please rephrase.

Line 16: Reciproc Blue (RB25), WaveOne Gold (WOG), EdgeOne Fire (EOF) and a recently introduced instrument OneRECI (OR), for a total of 40 new instruments. not a proper English please consider rephrasing.

Line 17. Before root canals shaping..... "canals???"

Line 56. Despite this, during the years, several limitations have been highlighted. Firstly, reciprocating instruments have been criticized for the possibility to extrude debris and bacteria in the periapical space in comparison to the continuous rotary instrumentation.  I suggest citing some papers here.

The discussion should explain more about the methodology employed in the current investigation and the authors should increase the references too. 

For a bright study like thing lacunae in the literature references should be avoided.

Reviewer 2 Report

Dear authors,

The work is well constructed and well presented. The SEM pictures are very explanatory, and the conclusion is very helpful.

The reciprocation of single files can offer a valuable clinical tool if used with caution, and thanks to research like the presented work, many dentists can avoid complications like file separation.

Only minor English mistakes need to be adjusted, for example

line 324 "instruments" should be "instrument"

line 326 "guarantees" should be "guarantee" and many other minor errors that better be corrected for the sake of perfection for your almost perfect work.
My best regards.

Reviewer 3 Report

The article is interesting, but I suggest the author make the explanation of the results extensive, it is fine to put a table and images, but they only serve as support, the first paragraph places the author: it is shown in the table, when the table only serves to give support to the text, that is, the author must explain extensively (text) in the results section, both the table and each of the images, in the same way he makes an image caption too long when he must place it in the form of text in the results section, you must improve it, so that it can be considered for publication,
